# Structure of a eukaryotic cholinephosphotransferase-1 reveals mechanisms of substrate recognition and catalysis

Lie Wang [1] ✉ & Ming Zhou [1] ✉

Phosphatidylcholine (PC) is the most abundant phospholipid in eukaryotic cell membranes. In eukaryotes, two highly homologous enzymes, cholinephosphotransferase-1 (CHPT1) and choline/ethanolamine phosphotransferase-1 (CEPT1) catalyze the final step of de novo PC synthesis. CHPT1/CEPT1 joins two substrates, cytidine diphosphate-choline (CDP-choline) and diacylglycerol (DAG), to produce PC, and $Mg^{2+}$ is required for the reaction. However, mechanisms of substrate recognition and catalysis remain unresolved. Here we report structures of a CHPT1 from *Xenopus laevis* (xlCHPT1) determined by cryo-electron microscopy to an overall resolution of ~3.2 Å. xlCHPT1 forms a homodimer, and each protomer has 10 transmembrane helices (TMs). The first 6 TMs carve out a cone-shaped enclosure in the membrane in which the catalysis occurs. The enclosure opens to the cytosolic side, where a CDP-choline and two $Mg^{2+}$ are coordinated. The structures identify a catalytic site unique to eukaryotic CHPT1/CEPT1 and suggest an entryway for DAG. The structures also reveal an internal pseudo two-fold symmetry between TM3-6 and TM7-10, and suggest that CHPT1/CEPT1 may have evolved from their distant prokaryotic ancestors through gene duplication.

Phosphatidylcholine (PC) is the main phospholipid in eukaryotic cell membranes, and serves as a precursor to a number of phospholipids and second messengers, such as phosphatidylserine, sphingomyelin, phosphatidic acid (PA), lyso-PC, and diacylglycerol (DAG)[1–5]. Inhibition of PC synthesis leads to arrested cell growth and apoptosis[6–8]. In eukaryotic cells, the production of PC is mainly achieved through de novo synthesis by the Kennedy pathway, which is a multi-stepped process and the last of which is the production of PC catalyzed by two highly homologous enzymes, cholinephosphotransferase-1 (CHPT1) and choline/ethanolamine phosphotransferase-1 (CEPT1)[9,10]. In CHPT1 and CEPT1, choline phosphate from CDP-choline is transferred to the free hydroxyl of a DAG to produce a PC (Supplementary Fig. 1,

ref. [11],[12]). Another closely related enzyme, ethanolamine phosphotransferase-1 (EPT1), catalyzes the transfer of ethanolamine phosphate from CDP-ethanolamine to DAG and produces phosphatidylethanolamine (PE) (Supplementary Fig. 1, ref. [13]). All three enzymes are found in the membranes of endoplasmic reticulum or Golgi complexes and critical for cell functions[14].

Although CHPT1, CEPT1, and EPT1 are phosphotransferases, they are classified into the superfamily of CDP-alcohol phosphatidyltransferases (CDP-AP, InterPro IPR000462), which are ubiquitous in both prokaryotic and eukaryotic organisms. CDP-APs share a signature sequence motif, $D_1xxD_2G_1xxAR…G_2xxxD_3xxxD_4$, and catalyze the displacement of cytidine monophosphate (CMP) from a CDP-linked

[1]Verna and Marrs McLean Department of Biochemistry and Molecular Biology, Baylor College of Medicine, Houston, TX, USA. ✉e-mail: liew@bcm.edu; mzhou@bcm.edu

alcohol with another alcohol. In prokaryotic cells, the CDP-linked alcohol is frequently a CDP-DAG and hence the name phosphatidyl-transferase. Structures of several bacterial CDP-APs have been reported[15–17], which define a common structural fold composed of six TMs and have the CDP-AP signature motif located on TM2 and TM3 that coordinate either one or two Mg$^{2+}$. The structures of eukaryotic CDP-APs remain elusive. Since most eukaryotic CDP-APs have 10 TMs and utilize different substrates from their prokaryotic relatives, we pursued their structure and examined their functions to gain a better understanding of the mechanisms.

## Results

### Function of CHPT1

Full-length *Xenopus laevis* CHPT1 (xlCHPT1, UniProt accession number Q4KLV1), which is ~68% identical and ~83% similar to both human CHPT1 and CEPT1, was expressed and purified (Methods, Supplementary Figs. 2a and 3). xlCHPT1 was purified as a homodimer and we examined its enzymatic activity and substrate preference. We first examined CDP-choline and 1,2-*sn*-diacylglycerol as substrates and measured the enzymatic activity by following the production of CMP (Methods and Supplementary Fig. 2b). The initial rate of CMP production at different concentrations of CDP-choline was determined and plotted in Fig. 1a, and the data are well-fit with a Michaelis-Menten equation with a $K_M$ of $18.0 \pm 1.9\ \mu M$ and a $V_{max}$ of $72.8 \pm 1.7$ nmol/min/mg. These values are similar to human CEPT1[12,18]. Next, we examined CDP-ethanolamine and 1,2-*sn*-diacylglycerol as substrates, and obtained a $K_M$ of $603.0 \pm 61.6\ \mu M$ and a $V_{max}$ of $55.6 \pm 1.6$ nmol/min/mg (Fig. 1b). Thus, xlCHPT1 has a clear preference for CDP-choline. We also examined the effect of divalent cations, and found that xlCHPT1 is active in the presence of either Mg$^{2+}$ or Mn$^{2+}$, but not in Ca$^{2+}$ or Zn$^{2+}$ (Fig. 1c). These functional properties of xlCHPT1, i.e., preference to CDP-choline and Mg$^{2+}$ over CDP-ethanolamine and Ca$^{2+}$, respectively, are consistent with previous reports of human CEPT1[12].

### Overall structure of xlCHPT1

We determined two structures of xlCHPT1 by single-particle cryo-electron microscopy (cryo-EM). The structure of xlCHPT1 in complex with CDP-choline and Mg$^{2+}$ has an overall resolution of ~3.7 Å, and that of xlCHPT1 in complex with CDP and Mg$^{2+}$, ~3.2 Å resolution (Fig. 2a, b, Supplementary Figs. 4–5). Both density maps are of sufficient quality to allow de novo model building of residues 20–383 (Supplementary Fig. 6a). The first 19 and the last 18 residues are not resolved, although they are predicted to form α-helices in a structural model generated by AlphaFold (Supplementary Fig. 7 and ref. 19). Since the main chain atoms of the two structures are almost identical, we will focus on the

structure of xlCHPT1 in complex with CDP and Mg$^{2+}$ in all the figures except in panels that show CDP-choline.

The structure of xlCHPT1 is a homodimer, and each xlCHPT1 protomer has 10 transmembrane helices. Based on the "positive-inside" rule[20], both the N- and C-terminus are located on the cytosolic side (Supplementary Fig. 6b). The first six TMs carve out a cone-shaped enclosure in the membrane with a prominent slit between TM5 and TM6. The wide end of the cone opens to the cytosolic side and the opening is partially covered by a well-defined two-stranded β-sheet, one strand from the N-terminus (β1) and another from the loop between TM4 and 5 (β2, Fig. 2c, d). The N-terminus preceding TM1, residues 20–62, is well-structured with three secondary structural elements, a short α-helix (α1), followed by a β-strand that forms part of the two-stranded β-sheet, and a long amphipathic helix (α2) that wraps around the perimeter of the cone. TMs7-10, which are present only in eukaryotic CDP-APs, form a structured bundle and are related to TMs3-6 by a pseudo twofold rotational symmetry (Supplementary Fig. 8). This internal repeat is not recognizable from the amino acid sequence, and we will discuss its significance in comparison to bacterial CDP-APs. Also located close to the wide end of the cone are the CDP-AP signature motif, the two Mg$^{2+}$, and one CDP-choline (Figs. 2c and 3a).

### Mg$^{2+}$ and CDP-choline-binding sites

Density for two Mg$^{2+}$ are prominent in both cryo-EM maps, and the Mg$^{2+}$ are coordinated by side chain carboxylates of Asp111, Asp114 from TM2 and Asp132, Asp136 from TM3 (Fig. 3b–d and Supplementary Fig. 6). The side chain carboxylates of Asp111 and Asp132 interact with both Mg$^{2+}$ simultaneously, forming two bidentate bridges. We define the Mg$^{2+}$ that interacts with Asp114 as Mg1 and the other one as Mg2 (Fig. 3d). The four aspartate residues are part of the conserved CDP-AP motif, and we validated their functional impact: mutating any of the aspartate to alanine reduces the enzymatic activity to <20% of that of the wild type xlCHPT1 (Fig. 3g and Supplementary Fig. 9). Densities for CDP or CDP-choline are clearly recognizable, with the cytidine ring coordinated by the side chain of Asn64 and by the backbone nitrogen of Ser124 (Fig. 3b, c, e, f). The diphosphate group in CDP is coordinated by Mg1, the side chain guanidinium group of Arg119, and the side chain hydroxyl group of Tyr34. The quaternary ammonium of CDP-choline is stabilized by the aromatic rings of Trp50 and Phe186 and by the side chain carboxylate of Glu43 (Fig. 3f). The importance of residues at the CDP-choline binding site is demonstrated by functional analysis: point mutations to these residues reduce enzymatic activity by 50–80% (Fig. 3g and Supplementary Fig. 9). Residues involved in the CDP-choline binding site are conserved in human homologs (Supplementary Fig. 3 and 10a–c).

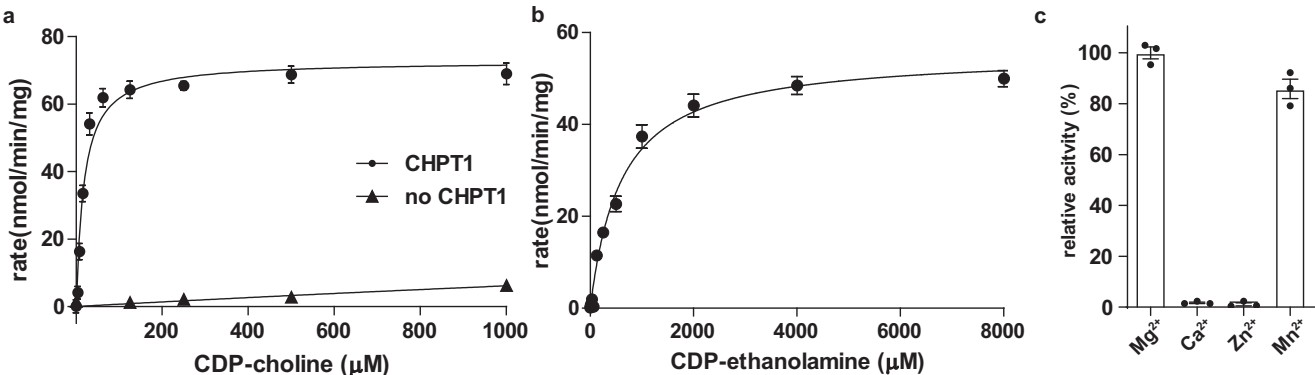

**Fig. 1 | Function of purified xlCHPT1. a–b** Initial rate of reaction in different concentrations of CDP-choline (**a**) or CDP-ethanolamine (**b**). **c** Relative activity of xlCHPT1 in the presence of different cations. Each symbol or bar is the average of three independent measurements, and error bars are standard errors of the mean (s.e.m.). The solid lines in **a**, **b** are fit of the data point to a Michaelis−Menten equation.

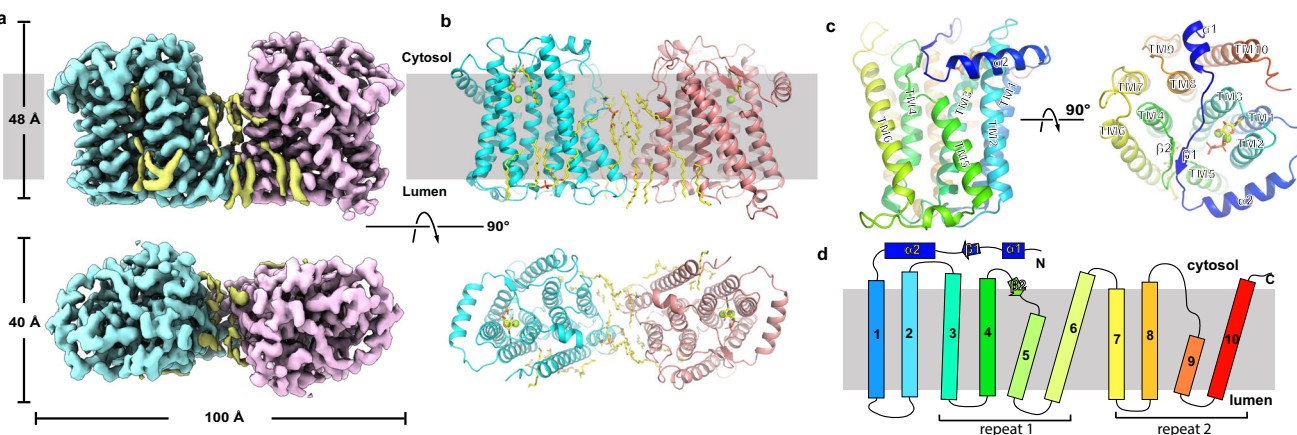

**Fig. 2 | Overall structure of xlCHPT1. a** Density map of dimeric xlCHPT1. Density of the two monomers is colored in cyan and pink, and the density of lipids is colored in yellow. **b** Model of dimeric xlCHPT1. Two monomers are shown as cartoons in cyan and pink, and substrates and lipids are shown as yellow sticks. Two $Mg^{2+}$ are shown as green spheres. **c** Structure of xlCHPT1 monomer in two orientations. **d** Membrane topology plot of an xlCHPT1 monomer.

**Fig. 3 | $Mg^{2+}$ and CDP-choline-binding sites. a** Overview of CDP-choline and $Mg^{2+}$-binding sites in two orientations. xlCHPT1 is shown as cartoon, CDP-choline as a stick, and $Mg^{2+}$ as green spheres. **b–c** Density maps of CDP or CDP-choline with $Mg^{2+}$. Density map is shown as transparent blue surface, $Mg^{2+}$ as spheres, and CDP or CDP-choline as sticks. **d–f** Coordination of $Mg^{2+}$, CDP, or CDP-choline in xlCHPT1. Binding site residues and CDP/CDP-choline are both shown as sticks but with a different color scheme. **g** Relative activity of wide type and mutants of xlCHPT1. Each symbol is an average of three independent measurements. Error bars are standard errors of the mean (s.e.m.). **h** The catalytic site, which is marked as a star, and the proposed catalytic residues, E129 and H133, are shown as sticks.

## Catalytic site and DAG entryway

Located in the vicinity of the bound $Mg^{2+}$ and CDP-choline, Glu129 and His133 on TM3 are highly conserved in eukaryotic CDP-AP enzymes but are absent in most of the bacterial CDP-APs (Fig. 3h, Supplementary Fig. 3 and 10d). The two residues do not have direct interactions with $Mg^{2+}$ or CDP-choline but His133 is ~7 Å away from the β-phosphate of CDP. A conserved histidine paired with glutamate or aspartate is commonly found in the active site of enzymes that catalyze

nucleophilic attack by a hydroxyl group, in which the histidine facilitates extraction of the proton from the hydroxyl group and the glutamate or aspartate stabilizes the protonated histidine[21–24]. We propose that Glu129 and His133 in xlCHPT1 are part of the catalytic center so that the free hydroxyl group of a DAG substrate is within hydrogen bond distance of His133. Mutations of His133 or Glu129 to alanine almost completely abolish the enzymatic activity of xlCHPT1 (Fig. 3g). To further explore this, we manually placed a 1, 2-sn-dioleoylglycerol into the enclosure with its hydroxyl group between His133 and the phosphate, and we noticed that the glycerol backbone of DAG could be coordinated by residues Asp136 and Gln175, with the two acyl chains extending through the slit between TM5 and 6 and thus partially buried in the hydrophobic core of the membrane (Supplementary Fig. 11). In both density maps, there are tubular shaped non-protein densities resembling two acyl chains at the entrance of the slit, suggesting that the slit is wide enough to accommodate two acyl chains (Supplementary Fig. 6c). In addition, TM6 does not pack tightly against the rest of the protein and thus could afford movement to allow unimpeded entry of DAG into the proposed catalytic site (Fig. 4a, b).

## Dimerization interface

xlCHPT1 forms a homodimer, and the dimer interface is mediated mainly by residues from TM7 and TM9 of each protomer (Fig. 4a, b). The aromatic rings of Phe336 and Tyr340 on TM9 form stacking interactions, and the hydroxyl group of Tyr340 could form a hydrogen bond with the side chain of Gln339 (Fig. 4c). In addition, Lys284 from TM7 could form a hydrogen bond with the side chain carbonyl of Asn342 from TM9. The dimer interface spans only ~10 Å at the luminal side of the membrane, leaving a large gap between the two monomers that are filled with lipids (Fig. 4d). TM9 is a short helix with three and half helical turns, and it is preceded by an extended amphipathic strand with its hydrophobic side packed against TM7 and TM8 and the hydrophilic side exposed to solvent. Because of this construction, the lipid bilayer at the dimer interface is significantly thinner with a hydrophobic core of ~25 Å (Fig. 4a and Supplementary Fig. 6b).

## Discussion

Here we report the structures of a eukaryotic CDP-AP, which reveal structural features that are not present in prokaryotic homologs. Interestingly, these features are predicted accurately by the AlphaFold (Supplementary Fig. 7a–d). The predicted structure deviates from the experimental structure with a root mean squared distance (r.m.s.d.) of 0.96 Å for Cα atoms. The deviations are mainly from the intracellular side of TM2 and TM3, incurred likely due to the presence of $Mg^{2+}$ in the experimental structure, and from the amphipathic helix, which may have intrinsic motions in the plain of the membrane (Supplementary Fig. 7e).

Several prokaryotic CDP-APs structures were reported previously, and all but one form homodimers[15–17,25–28]. When an xlCHPT1 monomer, which has 10 TMs, is aligned to a prokaryotic homolog, phosphatidyl inositol phosphate synthase from *Renibacterium salmoninarum* (*Rs*PIPS, pdb code 5D92), which has 6 TMs, TM1-3 of xlCHPT1 are aligned with that of *Rs*PIPS with a root mean squared distance (r.m.s.d.) of 1.9 Å, while TM4-6 and the amphipathic helix preceding TM1 have larger deviations, 3.1 Å, but follow a similar arrangement. Similar differences in alignments are observed when xlCHPT1 is aligned to other bacterial CDP-APs (Supplementary Fig. 12). In both eukaryotic and prokaryotic CDP-AP structures, the CDP-AP signature motif, $D_1xxD_2G_1xxAR...G_2xxD_3xxxD_4$, is located to TM2 and TM3. The spatial arrangement of these conserved residues is preserved between bacterial CDP-APs and xlCHPT1, and seems optimized for the coordination of two $Mg^{2+}$ and the CDP group. The structure of xlCHPT1 reveals that His133 and Glu129 may participate in the catalysis, and we propose that in xlCHPT1 and other eukaryotic CHPT1/CEPT1, His133 and Glu129 should be included in the signature motif, $D_1xxD_2G_1xxAR...G_2\underline{E}xxD_3\underline{H}xxD_4$, which facilitate extraction of a proton from DAG and thus nucleophilic attack on CDP-choline (Fig. 5c, Supplementary Fig. 3).

We speculate that the relatively smaller variations of TM1-3 between prokaryotic and eukaryotic CDP-APs are due to their common functional role in supporting the coordination of $Mg^{2+}$ and CDP group, while the larger deviations in the alignment of TM4-6 and the amphipathic helix and the additional two-stranded β sheet reflect adaptation of eukaryotic enzymes to different substrates. For

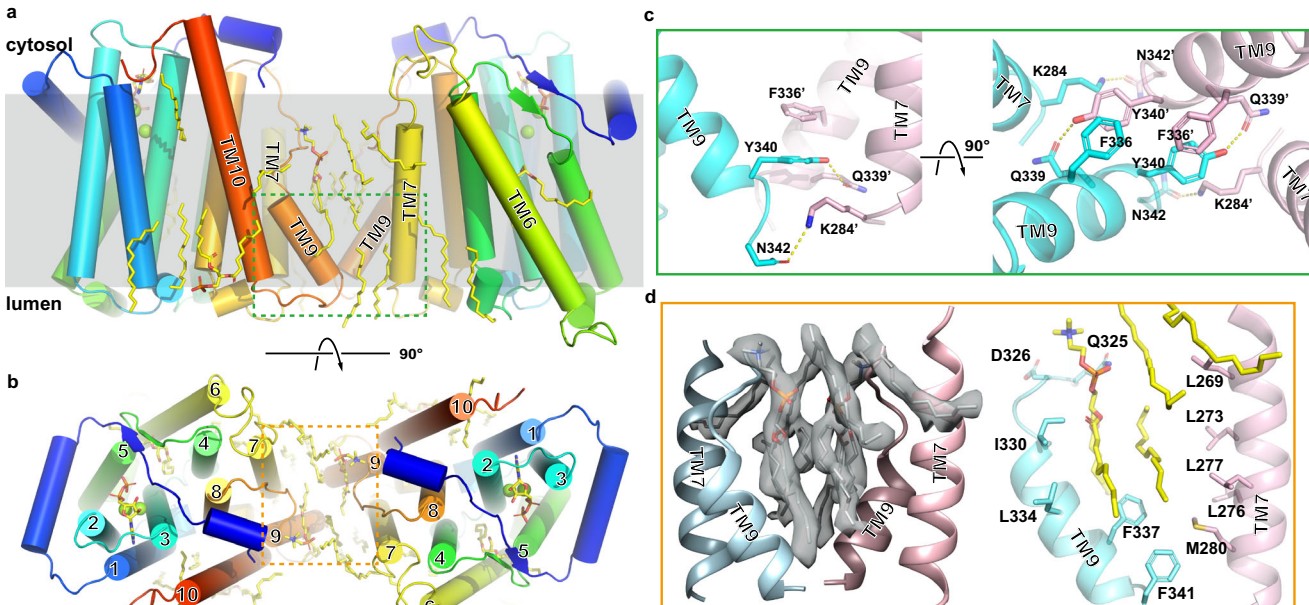

**Fig. 4 | Dimerization interface. a−b** Dimeric xlCHPT1 structure is shown as cartoon in two orientations, and lipids are shown as yellow sticks. The green and orange boxes mark the region with close-up views shown in **c** & **d**. **c** Residues at the dimer interface. Two monomers are shown in cyan and pink. **d** Lipids trapped at the dimer interface. Density of lipids is shown as a transparent gray surface, and partially resolved lipid molecules are shown as sticks.

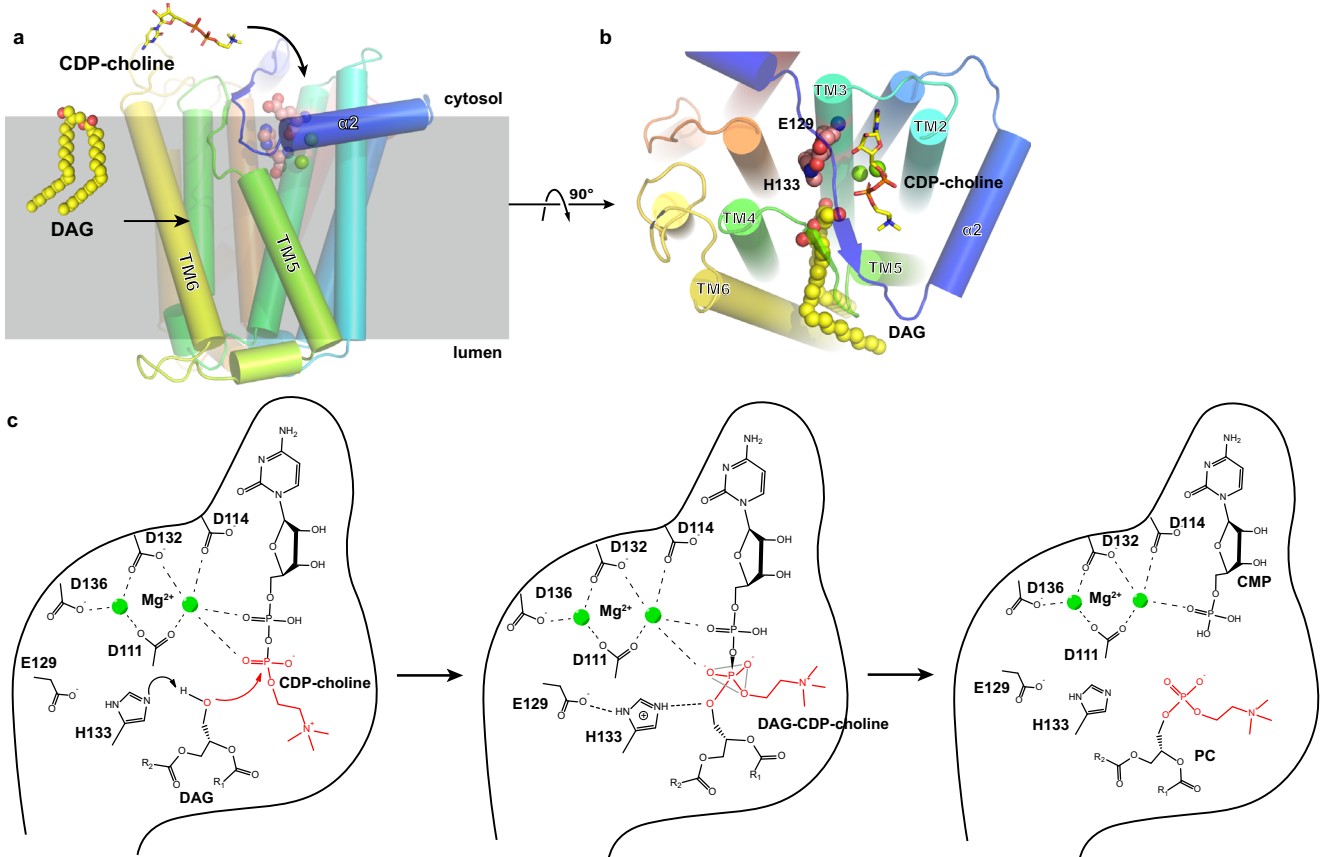

**Fig. 5 | Substrate entry and reaction mechanism. a–b** Proposed entry pathways for both substrates. CDP-choline and DAG are shown as stick and spheres, respectively. xlCHPT1 is shown as cartoon. Proposed active site residues H133 and E129 are shown as spheres. **c** Proposed catalytic mechanism of CHPT1. E129 and H133 activate the 3-hydroxyl on DAG to enhance nucleophilic attack on the phosphate group of CDP-choline.

example, *Rs*PIPS and *Sa*PgsA do not have the two-stranded β sheet, and the amphipathic helix wraps loosely around TMs 2 and 5 to allow a larger substrate, CDP-DAG, to enter the binding site (ref. 17,27), while the inositol phosphate substrate enters from the large cytosolic opening and is accommodated by additional positively charged residues at the cytosolic side of the enzyme (Supplementary Fig. 13b–d). In xlCHPT1, the two-stranded β-sheet partially covers the large cytosolic opening of the enzyme and seems to leave an entrance just enough for a CDP-choline or CDP-ethanolamine. In addition, the amphipathic helix wraps tightly around TM2 and TM5, and the two helices are more closely packed. However, there is a large slit between TM5 and TM6 that could allow lateral entrance of DAG from the cell membrane (Fig. 5a, b, Supplementary Fig. 13c).

We noticed that when an xlCHPT1 protomer is aligned to a dimeric prokaryotic CDP-AP, TM7-10 of xlCHPT1 align with TM3-6 of the neighboring protomer (Supplementary Fig. 8b). While this observation reinforces the notion that a core of six TMs is the minimum functional unit of a CDP-AP, it also provides an explanation for the internal pseudo-two-fold symmetry observed between TM3-6 and TM7-10 in xlCHPT1. It is likely that eukaryotic CDP-APs have evolved from prokaryotic ancestors by a gene duplication or fusion event followed by a loss of the first two TMs in the second unit. A search of InterPro database for CDP-APs domain architectures shows that a small fraction of bacterial CDP-APs are indeed composed of two tandem CDP-AP units (represented by InterPro B3T1S3).

In summary, structural and functional analysis of xlCHPT1 enhances our understanding of substrate recognition and catalysis in eukaryotic CDP-APs. Our study also establishes a framework for further investigations that will reveal the binding site of DAG, the

mechanisms of catalysis, and regulation of enzymatic activity by associated proteins and lipids.

## Methods

### Cloning, expression, and purification of *Xenopus laevis* CHPT1

The xlCHPT1 gene (NCBI accession number NP_001089575.1) was codon-optimized and cloned into the pFastBac-Dual expression vector for production of baculovirus[29]. The P3 virus was used to infect Hi Five (Trichoplusia ni) cells at a density of $3 \times 10^6$ cells/ml, then the cells were harvested after 48–60 h. Cell membranes were prepared by a hypotonic/hypertonic wash protocol as previously described[30]. Briefly, cells were lysed in a hypotonic buffer containing 10 mM 4-(2-hydroxyethyl)−1-piperazineethanesulfonic acid (HEPES) pH 7.5, 10 mM NaCl, and 2 mM β-mercaptoethanol (BME), 1 mM phenylmethylsulfonyl fluoride (PMSF), and 25 μg/ml DNase I. After ultracentrifugation at $55,000 \times g$ for 10 min, the pelleted cell membranes were resuspended in a hypertonic buffer containing 25 mM HEPES pH 7.5, 1 M NaCl, 2 mM BME, 1 mM PMSF, and 25 μg/ml DNase I, and were centrifuged again at $55,000 \times g$ for 20 min. The pelleted cell membranes were resuspended in 20 mM HEPES pH 7.5, 150 mM NaCl, 20% glycerol, and flash frozen in liquid nitrogen for further use.

Purified membranes were thawed and extracted in 20 mM HEPES pH 7.5, 150 mM NaCl, 2 mM BME, 4 mM $MgCl_2$, 1 pill of inhibitor cocktail tablet (Roche), and then solubilized with 1.5% (w/v) lauryl maltose neopentyl glycol (LMNG, Anatrace) at 4 °C for 2 h. After solubilization, cell debris was removed by centrifugation in $55,000 \times g$, 45 min, 4 °C, xlCHPT1 was purified from the supernatant with a cobalt-based affinity resin (Talon, Clontech). The C-terminal His6-tag was cleaved with tobacco etch virus protease at room temperature for 30 min. The protein was then concentrated to around 5 mg/ml

(Amicon 100 kDa cut-off, Millipore), and loaded onto a size-exclusion column (SRT-3C SEC-300, Sepax Technologies) equilibrated with 20 mM HEPES, 150 mM NaCl, 2 mM BME, 4 mM MgCl$_2$ and 0.1% (w/v) LMNG, 0.01 (w/v) cholesteryl hemisuccinate (CHS, sigma). For the sample used in cryo-EM, the size-exclusion column was equilibrated with 20 mM HEPES, 150 mM NaCl, 2 mM BME, 4 mM MgCl$_2$, and 0.02% (w/v) GDN. Ligands were added after the size-exclusion chromatography. For the xlCHPT1-Mg$^{2+}$-CDP-choline sample, 10 mM CDP-choline was added. For the xlCHPT1-Mg$^{2+}$-CDP sample, 10 mM CDP and 0.5 mM DAG were added.

xlCHPT1 mutants were generated using the QuikChange method (Stratagene) with primers (Supplementary Table 2) and the entire cDNA was sequenced to verify the mutation. Mutants were expressed and purified following the same protocol as wild type.

### Cloning, expression, and purification of yeast pyrimidine 5′-nucleotidase SDT1

The yeast SDT1 gene (NCBI accession number NM_001181089.1) was codon-optimized and cloned into the pMCSG28 vector containing a C-terminal His-tag for expression. The expression protocol is similar to the previously described[31]. Transformed BL21 *E.coli* cells were grown in Luria-Bertani (LB) media until reaching an optical density of 0.6–0.8 and then induced with 0.5 mM IPTG. Cells were then grown at 20 °C overnight, spun down at $4000 \times g$ for 15 min, and resuspended in a solution of 20 mM HEPES pH 7.5, 150 mM NaCl, and 10% glycerol. To purify SDT1, cell pellet in 20 mM HEPES pH 7.5, 150 mM NaCl, and 10% glycerol with cocktail inhibitors was sonicated on ice and centrifuged in 55,000 rpm for 30 min at 4 °C. SDT1 with his-tag was purified from the supernatant with a cobalt-based affinity resin (Talon, Clontech). The eluted protein was then concentrated to around 5 mg/ml (Amicon 50 kDa cutoff, Millipore), and loaded onto a size-exclusion column (SRT-3C SEC-300, Sepax Technologies) equilibrated with 20 mM HEPES, 150 mM NaCl, 2 mM BME, 4 mM MgCl$_2$. The purified SDT1 protein was concentrated to -1 mg/mL, then stored at −80 °C after aliquoted and fast frozen in liquid nitrogen.

### Cryo-EM sample preparation and data collection

Cryo grids were prepared on the Thermo Fisher Vitrobot Mark IV. Quantifoil R1.2/1.3 Cu grids were glow-discharged using the Pelco Easyglow. Concentrated xlCHPT1 (3.5 µL) was applied to glow-discharged grids. After blotting with filter paper (Ted Pella) for 3.5–4.5 s, the grids were plunged into liquid ethane cooled with liquid nitrogen. For cryo-EM data collection, movie stacks were collected using EPU (Thermo Fisher Scientific) on a Titan Krios at 300 kV with a Quantum energy filter (Gatan), at a nominal magnification of ×81,000 and with defocus values of −2.0 to −0.8 µm. A K3 Summit direct electron detector (Gatan) was paired with the microscope. Each stack was collected in the super-resolution mode with an exposing time of 0.175 s per frame for a total of 50 frames. The dose was about 50 e⁻ per Å$^2$ for each stack. The stacks were motion-corrected with Relion 3 and binned ($2 \times 2$) so that the pixel size was 1.07 Å[32]. Dose weighting was performed during motion correction, and the defocus values were estimated with Gctf[33].

### Cryo-EM data processing

For xlCHPT1-Mg$^{2+}$-CDP data set, a total of 5,734,696 particles were automatically picked in RELION 3.1 with template picking from 8100 images[34], and imported into cryoSPARC[35]. After three rounds of two-dimensional (2D) classifications, 21 classes (containing 734,414 particles) were selected out of 200 2D classes for ab initio three-dimensional (3D) reconstruction, which produced one good class with recognizable structural features and three bad classes that did not have structural features. Both the good and bad classes were used as references in three rounds of heterogeneous refinement (cryoSPARC)

and yielded a good class at 4.06 Å from 465,752 particles. Then non-uniform refinement (cryoSPARC) was performed with C2 symmetry and an adaptive solvent mask, CTF refinement yielded a map with an overall resolution of 3.2 Å.

For xlCHPT1-Mg$^{2+}$-CDP-choline data set, a total of 3,859,534 particles were automatically picked in RELION 3.1 with template picking from 5510 images and imported into cryoSPARC. After three rounds of 2D classification, 15 classes (containing 1,229,270 particles) were selected out of 200 2D classes for ab initio three-dimensional 3D reconstruction, which produced one good class with recognizable structural features and three bad classes that did not have structural features. Both the good and bad classes were used as references in three rounds of the heterogeneous refinement (cryoSPARC) and yielded a good class at 4.15 Å from 381,720 particles. Then nonuniform refinement (cryoSPARC) was performed with C2 symmetry and an adaptive solvent mask, CTF refinement yielded a map with an overall resolution of 3.68 Å. Resolutions were estimated using the gold-standard Fourier shell correlation with a 0.143 cutoff[36] and high-resolution noise substitution[37]. Local resolution was estimated using ResMap[38].

### Model building and refinement

The structural models of xlCHPT1 were built de novo into the density map starting with poly-alanine, and sidechains were then added onto the model based on the map. Model building was conducted in Coot[39]. Structural refinements were carried out in PHENIX in real space with secondary structure and geometry restraints[40]. The EMRinger Score was calculated as described[41].

### Enzyme-coupled enzymatic assay

xlCHPT1 activity was measured using an absorbance-based coupled-enzyme assay (Supplementary Fig. 2b). All reaction assays were done in a buffer with 20 mM HEPES, pH 7.5, 150 mM NaCl, 0.02% GDN, 5 mM MgCl$_2$. In single point activity assay, final concentrations of CDP-choline and 1,2-*sn*-diacylglycerol were 0.2 mM and 0.25 mM, respectively. Reactions were initiated with the addition of 10 µg purified xlCHPT1 protein. After reaction at 37 °C for 15 mins, reactions were stopped by heating up at 95 °C for 5 min. After removing the precipitated protein with centrifugation, purified yeast pyrimidine 5′-nucleosidase SDT1 was added into the supernatant, then react at 37 °C for 20 mins. Finally, phosphate dye (Sigma, MAK030) was added to the reaction solution and A$_{650 nm}$ was measured in 96 plates. When CDP-choline or CDP-ethanolamine concentrations are varied, DAG concentration is fixed at 250 µM. The initial rate versus different concentrations of CDP-choline and CDP-ethanolamine can be fit with a Michaelis–Menten equation.

### Reporting summary

Further information on research design is available in the Nature Portfolio Reporting Summary linked to this article.

## Data availability

The atomic coordinates of xlCHPT1 in complex with CDP-choline have been deposited in the PDB (http://www.rcsb.org) under the accession code 8ERP. The atomic coordinates of xlCHPT1 in complex with CDP have been deposited in the PDB under the accession code 8ERO. Their corresponding electron microscopy maps have been deposited in the Electron Microscopy Data Bank (https://www.ebi.ac.uk/pdbe/emdb/) under the accession codes EMD-28557 and EMD-28556, respectively. Source data are provided with this paper.

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

## Acknowledgements

This work was supported by grants from NIH (DK122784 and GM145416 to M.Z.). We acknowledge the cryo-EM cores at Baylor College of Medicine (CPRIT Core Facility Award RP190602) and the University of Texas Health Science Center in Houston for their support in grid preparation and screening. We are grateful to the Laboratory for BioMolecular Structure (LBMS) supported by the DOE Office of Biological and Environmental Research (KP1607011) for the support in data collection.

## Author contributions

M.Z. and L.W. conceived and designed the project. L.W. performed biochemistry, cryo-EM data collection and processing, and model building; M.Z. and L.W. analyzed the data and wrote the manuscript.

## Competing interests

The authors declare no competing interests.
