## [Peer Review File · Nature Communications]

REVIEWER COMMENTS

Reviewer #1 (Remarks to the Author):

In this work the integral membrane *Xenopus* CHPT1 protein structure was determined by cryoEM to a resolution of 3.2-3.6 Angstroms. Human CHPT1 has been determined to synthesize the major membrane phospholipid phosphatidylcholine (PC) almost exclusively and is localized to the Golgi. Two other similar enzymes have been identified in humans, CEPT1 which can synthesize PC and phosphatidylethanolamine (PE) in the ER, and a PE specific EPT1.

Xenopus CHPT1-His6x was over-expressed and then purified from baculovirus using Co²⁺ resin, excision of the 6x-His tag, and then size exclusion chromatography prior to cryoEM. The structure was solved in the presence of the substrate CDP-choline as well as with CDP and the required cofactor Mg²⁺. The purified protein displayed a capacity to synthesize both PC and PE in an in vitro enzyme assay, with a strong preference for PC. The various CHPT1 variants produced were purified using a similar protocol and structure-activity relationships were probed by modelling prediction, site-directed mutagenesis, and enzyme activity assays. Substrate binding and catalytic residues were identified, including probing the CDP-aminoalcohol catalytic motif. This is the first experimentally resolved structure for a CHPT1 and many new molecular aspects by which the major membrane phospholipid PC (i.e. the synthesis of eukaryotic membranes) were described.

Comments

1. I would suggest a paragraph/figure on how the cryoEM determined structure compares to that predicted by AlphaFold2
2. On page 3 (Function of CHPT1) the first few sentences make it unclear if the CHPT1 from *Xenopus* is 68% identical and 83% similar to both CHPT1 and CEPT1. As these are different genes I would separate and discuss percent similarity to human CHPT1 and human CEPT1 in separate sentences to clarify.
3. Methods (Cloning, expression and purification of *Xenopus laevis* CHPT1) – “codon” should be “codon”
4. Methods (Cloning, expression and purification of yeast pyrimidine...) – again, “codon” should be “codon”
5. Methods (Model building and refinement) – I believe “poly-alanine” should be “poly-alanine”
6. I might suggest a title with more gravitas, perhaps along the lines of “Determining how phosphatidylcholine is synthesized by structure and mechanistic studies of a eukaryotic cholinephosphotransferase”

Reviewer #2 (Remarks to the Author):

In this manuscript, the authors solve the first structure of a eukaryotic CHPT1, which carries out the final step of the Kennedy pathway of phosphatidylcholine biosynthesis. This is clearly an important target and is also a member of the CDP-alcohol superfamily of phosphatidyltransferases. The eukaryotic members have a different architecture than their prokaryotic counterparts and thus this work fills up an important void.

Key findings

The authors present two cryoEM structures of *Xenopus laevis* CHPT1, with CDP and CDP-choline bound in the active sites along with two Mg²⁺ ions. Although structures of related prokaryotic enzymes had been solved, the new structures reveal a novel architecture and enable the authors to propose a structure based mechanism of catalysis.

The authors have then done extensive mutational analysis of proposed functionally

important residues demonstrating the expected effect on enzymatic activity.

From their structures, the authors have also identified two residues, H133 and E129, hitherto not proposed as part of the signature motif. From docking, biochemical and sequence conservation, they propose these two be included in the signature motif for eukaryotic members of this superfamily.

The manuscript is written clearly and all the experiments have been planned carefully. The observations and conclusions are in agreement with the experimental data.

This manuscript in its current form requires a few points to be addressed before it can be considered for publication in Nature Comm.

Critique

1. One aspect that the authors dwell on at length is the dimerization. The dimerization surface seems to be fairly small and it is not clear if dimerization is -1) needed for biochemical activity and 2) is an artifact of surface delipidation during detergent extraction and purification. Although LMNG is not known to be a harsh detergent, such adventitious dimerization has been reported in the literature (see PNAS 103, 1723–1726 (2006)). So the following experiments would be important

Does the protein purify as a dimer in the presence of phospholipids (0.1 mg/ml.) added to the detergent in the buffer at the metal affinity and size-exclusion step ?

Can the dimer be broken by mutation of Y340 and K284 and thus disrupting the contacts ?

In the above cases, if the enzyme does indeed form a monomer, is it functionally active ?

2. Now that there are AlphaFold predicted structures of this enzymes, how does it compare with the experimental structure ? This is not to discredit the authors' structures, that contain substrates that are not in the AlphaFold prediction. However, this will be an important piece of information and should be included in the supplementary figures.

3. Although the authors discover and establish the importance of H233 and E129, they haven't proposed a unifying mechanistic framework that integrates the need for this departure from the prokaryotic enzymes. A discussion on this would enrich the manuscript. For example, could this have something to do with that the substrate alcohol in this case also has two long chain acyl groups and thus is constrained to be in the membrane, thereby imposing constraints on the relative positioning of the reactive groups ?

4. It seems that among the prokaryotic counterparts, the most similarity is with PgsA. In fact, the proposal in Fig. 5a is quite similar to the proposal in the graphical abstract in Curr. Res. Struct. Biol. 3, 312–323 (2021). So, it would be useful to have a similar analysis comparing with PgsA as the authors have done for RsPIPS and the figure included in the supplementary material.

Minor queries

1. Fig 3b,c : It would be useful to add sidechain residue numbers.

2. Supplementary Fig.12d. Typo in figure caption. Please omit xICHPT1 from ".....and RsPIPS xICHPT1".

3. The methodology section on cryoEM data processing could be improved by including the

data processing parameters in various steps as well as adding them in the data processing scheme in supplementary figure section where appropriate.

4. Are there any differences between maps generated from C1 and C2 processed particles with respect to the lipid densities given the effects of symmetry averaging?

POINT-BY-POINT RESPONSE TO REVIEWER COMMENTS

Reviewer #1 (Remarks to the Author):

In this work the integral membrane *Xenopus* CHPT1 protein structure was determined by cryoEM to a resolution of 3.2-3.6 Angstroms. Human CHPT1 has been determined to synthesize the major membrane phospholipid phosphatidylcholine (PC) almost exclusively and is localized to the Golgi. Two other similar enzymes have been identified in humans, CEPT1 which can synthesize PC and phosphatidylethanolamine (PE) in the ER, and a PE specific EPT1.

Xenopus CHPT1-His6x was over-expressed and then purified from baculovirus using Co²⁺ resin, excision of the 6x-His tag, and then size exclusion chromatography prior to cryoEM. The structure was solved in the presence of the substrate CDP-choline as well as with CDP and the required cofactor Mg²⁺. The purified protein displayed a capacity to synthesize both PC and PE in an in vitro enzyme assay, with a strong preference for PC. The various CHPT1 variants produced were purified using a similar protocol and structure-activity relationships were probed by modelling prediction, site-directed mutagenesis, and enzyme activity assays. Substrate binding and catalytic residues were identified, including probing the CDP-aminoalcohol catalytic motif. This is the first experimentally resolved structure for a CHPT1 and many new molecular aspects by which the major membrane phospholipid PC (i.e. the synthesis of eukaryotic membranes) were described.

Comments

1. I would suggest a paragraph/figure on how the cryoEM determined structure compares to that predicted by AlphaFold2

AlphaFold2 does a very good job predicting the structure, including features that are absent in the structures of the bacterial homologs, such as the parallel beta sheet on the intracellular side and the location of the amphipathic helix. There are only minor differences between the prediction and the structure. We added the model alignment in supplementary figure 7 and made a comment in Discussion.

2. On page 3 (Function of CHPT1) the first few sentences make it unclear if the CHPT1 from *Xenopus* is

68% identical and 83% similar to both CHPT1 and CEPT1. As these are different genes I would separate and discuss percent similarity to human CHPT1 and human CEPT1 in separate sentences to clarify.

Xenopus CHPT1 is 68% identical to both human CHPT1 and human CEPT1, and is 83% similar to both human CHPT1 and human CEPT1. We modified the sentence in question for clarity.

3. Methods (Cloning, expression and purification of *Xenopus laevis* CHPT1) – “condon” should be “codon”

Fixed

4. Methods (Cloning, expression and purification of yeast pyrimidine...) – again, “condon” should be “codon”

Fixed

5. Methods (Model building and refinement) – I believe “poy-alanine” should be “poly-alanine”

Fixed

6. I might suggest a title with more gravitas, perhaps along the lines of “Determining how phosphatidylcholine is synthesized by structure and mechanistic studies of a eukaryotic cholinephosphotransferase”

We changed the title to “Structure of a eukaryotic cholinephosphotransferase-1 reveals mechanisms of substrate recognition and catalysis”.

Reviewer #2 (Remarks to the Author):

In this manuscript, the authors solve the first structure of a eukaryotic CHPT1, which carries out the final step of the Kennedy pathway of phosphatidylcholine biosynthesis. This is clearly an important target and is also a member of the CDP-alcohol superfamily of phosphatidyltransferases. The eukaryotic members have a different architecture than their prokaryotic counterparts and thus this work fills up an important void.

Key findings

The authors present two cryoEM structures of *Xenopus laevis* CHPT1, with CDP and CDP-choline bound in the active sites along with two Mg²⁺ ions. Although structures of related prokaryotic enzymes had been solved, the new structures reveal a novel architecture and enable the authors to propose a structure based mechanism of catalysis.

The authors have then done extensive mutational analysis of proposed functionally important residues demonstrating the expected effect on enzymatic activity.

From their structures, the authors have also identified two residues, H133 and E129, hitherto not proposed

as part of the signature motif. From docking, biochemical and sequence conservation, they propose these two be included in the signature motif for eukaryotic members of this superfamily.

The manuscript is written clearly and all the experiments have been planned carefully. The observations and conclusions are in agreement with the experimental data.

This manuscript in its current form requires a few points to be addressed before it can be considered for publication in Nature Comm.

Critique

1. One aspect that the authors dwell on at length is the dimerization. The dimerization surface seems to be fairly small and it is not clear if dimerization is -1) needed for biochemical activity and 2) is an artifact of surface delipidation during detergent extraction and purification. Although LMNG is not known to be a harsh detergent, such adventitious dimerization has been reported in the literature (see PNAS 103, 1723–1726 (2006)). So the following experiments would be important

Does the protein purify as a dimer in the presence of phospholipids (0.1 mg/ml.) added to the detergent in the buffer at the metal affinity and size-exclusion step ?

Can the dimer be broken by mutation of Y340 and K284 and thus disrupting the contacts ?

In the above cases, if the enzyme does indeed form a monomer, is it functionally active ?

We appreciate the comment on dimerization, and we are able to address the question posed by the Reviewer. xICHPT1 remains as a dimer in LMNG with or without addition of phospholipids during the purification process. However, we observed predominantly monomeric xICHPT1 when xICHPT1 is purified in n-Decyl-beta-Maltoside. The two FPLC profiles shown below are from the same column. We also found that the monomeric xICHPT1 does not have significant enzymatic activity. We decided not to include these data in the current manuscript, because we feel that the tenuous protein contact and the significantly thinner bilayer at the dimer interface are worthy of a more rigorous and detailed follow-up study.

2. Now that there are AlphaFold predicted structures of this enzymes, how does it compare with the

experimental structure ? This is not to discredit the authors' structures, that contain substrates that are not in the AlphaFold prediction. However, this will be an important piece of information and should be included in the supplementary figures.

We agree and added a supplementary figure.

3. Although the authors discover and establish the importance of H233 and E129, they haven't proposed a unifying mechanistic framework that integrates the need for this departure from the prokaryotic enzymes. A discussion on this would enrich the manuscript. For example, could this have something to do with that the substrate alcohol in this case also has two long chain acyl groups and thus is constrained to be in the membrane, thereby imposing constraints on the relative positioning of the reactive groups ?

We appreciate and agree with reviewer's insightful comments. Bacterial homologs, including PIPS and PgsA, lack the equivalent of H233 and E129, which are prevalent in eukaryotic CDP-APs. In bacterial homologs, the nucleophilic attacking group is a hydroxyl from a phosphate, while in eukaryotic CDP-APs, it is the hydroxyl from DAG. We surmise that the hydroxyl group from DAG is better activated by the coordinated influence from H233 and E129, while the hydroxyl from phosphate may be activated by the aspartate residue that also coordinate the Mg^{2+} ion. Thus eukaryotic CDP-APs evolved to achieve more efficient nucleophilic attack facilitated by the H233/E129 pair. This hypothesis will have to be tested experimentally, and we included these thoughts in the main text.

4. It seems that among the prokaryotic counterparts, the most similarity is with PgsA. In fact, the proposal in Fig. 5a is quite similar to the proposal in the graphical abstract in Curr. Res. Struct. Biol. 3, 312–323 (2021). So, it would be useful to have a similar analysis comparing with PgsA as the authors have done for RsPIPS and the figure included in the supplementary material.

We agree and included PgsA in the supplementary figure 13.

Minor queries

1. Fig 3b,c : It would be useful to add sidechain residue numbers.

Fixed

2. Supplementary Fig.12d. Typo in figure caption. Please omit xlCHPT1 from “.....and RsPIPS xlCHPT1”.

Fixed

3. The methodology section on cryoEM data processing could be improved by including the data processing parameters in various steps as well as adding them in the data processing scheme in supplementary figure section where appropriate.

We added some more details in the method section.

4. Are there any differences between maps generated from C1 and C2 processed particles with respect to the lipid densities given the effects of symmetry averaging?

Lipid densities are present and overall similar in both C1 and C2 symmetry maps. Not surprisingly, the map constrained with C2 symmetry has stronger lipid densities.

Fig:

REVIEWERS' COMMENTS

Reviewer #2 (Remarks to the Author):

The authors have addressed all comments. Recommend publication